# Analytical Validation of an Immunohistochemical 7-Biomarker Prognostic Assay (immunoprint^®^) for Early-Stage Cutaneous Melanoma in Archival Tissue of Patients with AJCC v8 T2–T3 Disease

**DOI:** 10.3390/diagnostics13193096

**Published:** 2023-09-29

**Authors:** Mirjana Ziemer, Beate Weidenthaler-Barth, Philipp Gussek, Maja Pfeiffer, Johannes Kleemann, Katrin Bankov, Peter J. Wild, Silke Seibold, Priyavathi Sureshkumar, Patricia Nickel, Anton Strobel, Markus Werner, Stephan Grabbe

**Affiliations:** 1Department of Dermatology, Allergology and Venereology, University Medical Center, 04103 Leipzig, Germany; philipp.gussek@medizin.uni-leipzig.de; 2Department of Dermatology, University Medical Center Mainz, 55131 Mainz, Germany; beate.weidenthaler-barth@unimedizin-mainz.de (B.W.-B.); mpfeif@students.uni-mainz.de (M.P.); stephan.grabbe@unimedizin-mainz.de (S.G.); 3Department of Dermatology, Venerology and Allergology, University Hospital Frankfurt, Goethe University, 60590 Frankfurt am Main, Germany; johannes.kleemann@kgu.de; 4Dr. Senckenberg Institute of Pathology, University Hospital Frankfurt, Goethe University, 60590 Frankfurt am Main, Germany; katrin.bankov@kgu.de (K.B.); peter.wild@kgu.de (P.J.W.); 5Frankfurt Institute for Advanced Studies, 60438 Frankfurt am Main, Germany; 6Synvie GmbH, 80992 Munich, Germany; silke.seibold@synvie.de (S.S.); priyavathi.sureshkumar@synvie.de (P.S.); patricia.nickel@synvie.de (P.N.); anton.strobel@synvie.de (A.S.); markus.werner@synvie.de (M.W.)

**Keywords:** cutaneous melanoma, 7-marker signature prognostic assay (immunoprint^®^), risk stratification, immunohistochemistry, analytical validation, reproducibility, repeatability, reliability, concordance

## Abstract

Selected patients with early-stage melanoma have a “hidden high risk” of poor oncologic outcomes. They might benefit from clinical trials, and ultimately, if warranted by trial results, judicious everyday use of adjuvant therapy. A promising tool to identify these individuals is the immunoprint^®^ assay. This immunohistochemical 7-biomarker prognostic test was clinically validated in three independent cohorts (N = 762) to classify early-stage patients as high-risk or low-risk regarding melanoma recurrence and mortality. Using College of American Pathologists (CAP) recommendations, we analytically validated this assay in primary melanoma specimens (N = 20 patients). We assessed assay precision by determining consistency of risk classification under repeated identical conditions (repeatability) or across varying conditions (reproducibility), involving separate assay runs, operators (laboratory scientists), and/or observers (e.g., dermatopathologists). Reference classification was followed by five analytical validation phases: intra-run/intra-operator, intra-observer, inter-run, inter-operator, and inter-observer. Concordance of classifications in each phase was assessed via Fleiss’ kappa (primary endpoint) and percent agreement (secondary endpoint). Seven-marker signature classification demonstrated high consistency across validation categories (Fleiss’ kappa 0.864–1.000; overall percent agreement 95–100%), in 9/10 cases, exceeding, and in 1/10 cases, closely approaching, CAP’s recommended 0.9 level. The 7-marker assay has now been verified to provide excellent repeatability, reproducibility, and precision, besides having been clinically validated.

## 1. Introduction

Increasing evidence suggests that the putatively low-risk American Joint Committee on Cancer version 8 (AJCC v8) early stages of cutaneous melanoma (stages I–IIA) actually harbor appreciable numbers of patients at high-risk of recurrence and melanoma-specific mortality [1,2,3,4,5,6,7,8]. This high-risk subgroup might benefit from intensified post-surgical surveillance, and these individuals might be candidates for clinical trials of adjuvant therapy. Indeed, if clinical trials demonstrate superior outcomes with these strategies, such interventions potentially could become part of everyday practice. The benefit of adjuvant therapy was recently shown in patients with AJCC v8 stage IIB or IIC melanoma [9], leading to U.S. and E.U. regulatory approval of pembrolizumab in these settings. Additionally, preliminary analysis of the Phase III CheckMate76K trial has suggested recurrence-free survival (RFS) benefits of adjuvant nivolumab versus placebo [10]. This observation has led to applications for U.S. and E.U. regulatory approval of this modality in the stage IIB/IIC setting [11].

A promising tool to detect “hidden high-risk patients” among those with AJCC v8 early-stage melanoma is an immunohistochemistry-based 7-biomarker prognostic assay (immunoprint^®^, Synvie GmbH, Munich, Germany). This test also has the potential to increase confidence regarding the status of truly low-risk patients, helping spare them the expense and side effects of unneeded treatment.

The 7-marker signature assay dichotomizes patients into a high-risk classification or a low-risk classification (Figure 1) regarding disease-free survival/RFS, distant metastasis-free survival, melanoma-specific survival (MSS), and/or overall survival [12,13,14]. It does so by generating a total score based on the presence in primary melanoma tissue of five “risk” markers (Bax, Bcl-X, COX-2, PTEN, and CD20-positive B-lymphocytes) and two “protective” markers (loss of MTAP and of ß-catenin) [12].

Notably, in a large (N = 439) study by Meyer et al. [13], 7-marker signature high-risk classification could identify stage I–IIA patients with a 7.5-year (MSS) resembling 10-year MSS in stage IIB or stage IIC patients in two cohorts studied by Garbe et al. [4] or in the AJCC v8 cohort [15]. In the Meyer et al. study [13], 7.5-year MSS was 74.3% (95% confidence interval [CI] 62.1–88.8%) in stage IA–IIA patients classified as high-risk by the 7-marker signature assay, and 71.8% (95% CI 58.6–88.1%) in stage IB–IIA subgroup members with the high-risk classification. In comparison, 10-year MSS was 72.0–79.9% in stage IIB patients and 57.6–64.7% in stage IIC patients in the Garbe et al. cohorts [4], and 82% and 75%, respectively, in such patients in the AJCC v8 cohort [15].

To date, clinical validation of 7-marker signature assay risk data as a significant independent predictor of melanoma survival outcomes has been reported in a total of 762 patients from three independent cohorts [12,13,14]. Such validation has taken place both retrospectively [12,14] and using prospectively-collected specimens [13], in the presence [14] and absence [12] of sentinel lymph node staging, and applying both tissue microarrays [12] and the whole-tissue melanoma sections [13,14] more typically encountered in clinical routine. Of interest, in the recent Meyer et al. study [13], the addition of 7-marker signature assay risk data to key clinicopathological and demographic characteristics, including factors integral to AJCC staging, markedly improved prognostic performance compared to use of the characteristics alone.

However, although clinical validation is critical to establish the utility of an assay in clinical trials, and, potentially, in everyday practice, analytical validation of the assay is also essential [16]. Analytical validation is achieved by demonstrating the test’s precision, i.e., repeatability and reproducibility. Repeatability is shown by the assay generating closely corresponding results in repeated measurements made under identical conditions, e.g., among multiple runs, when performed by a given operator (laboratory scientist), or when scored by a given observer (e.g., dermatopathologist). Reproducibility is shown by the test generating closely corresponding results in measurements made across varying conditions, e.g., when assay runs are carried out by different operators or when classifications are made by different observers.

Guidelines issued by the College of American Pathologists (CAP) [17] and the Clinical Laboratory Standards Institute (CLSI) [18] provide well-accepted recommendations for the analytical validation of immunohistochemistry-based tests. For this reason, we performed the present study to analytically validate the 7-marker signature assay according to these recommendations. We focused on the signature’s binary risk classification, i.e., high-risk or low-risk, which is the assay’s key output used to help inform clinical decision making. As reported below, through comprehensive analysis of samples from 20 patients, we could verify the high precision, repeatability, and reproducibility of the 7-marker signature assay, and thereby, could analytically validate this test.

## 2. Materials and Methods

### 2.1. Study Design

This was a multicenter study, with a pre-specified protocol and statistical analyses, that retrospectively assessed archival formalin-fixed, paraffin-embedded (FFPE) primary melanoma tissue samples. These samples were collected at three academic tertiary medical centers in Germany, and were not included in prior published studies regarding 7-marker signature testing [12,13,14].

Study assays were performed at a central laboratory, that of Synvie GmbH in Munich, Germany. Per protocol, and to help determine study eligibility, the only data regarding the specimens that the study centers provided to the laboratory and study staff or to the study biostatisticians were the AJCC v8 T-stage, the original tumor thickness, and the date of the initial cutaneous melanoma diagnosis; all these data were provided in pseudonymized form. Additional histopathological data, demographic or clinical information, or outcome data were not conveyed by the study centers. Observers were blinded to fellow observers’ evaluations as well as to their own previous ratings. Each observer’s evaluation of each run was documented on a new, blank laboratory journal page, and could not be subsequently consulted.

For all eligible specimens, a “reference staining” for the 7 biomarkers in the assay was performed first, to determine the sample’s risk category. The “reference risk classification” enabled a selection of appropriate numbers of high-risk and low-risk samples for inclusion into the overall study sample and into each of the study’s two primary tumor stage subgroups (T2 and T3). The reference risk classification, made before the assay validation was performed, served as a comparator for all risk categorizations made in each assay validation phase.

In line with CAP [17] and CLSI [18] recommendations, the study had five validation phases (Figure 2 and Figure 3, Table 1): (A) intra-run/intra-operator, (B) intra-observer, (C) inter-run, (D) inter-operator, and (E) inter-observer. The intra-run/intra-operator phase involved three assay runs being conducted by a single operator on the same day, and risk categorization being performed by a single observer. The intra-observer phase entailed one assay run being conducted by a single operator. Seven-marker signature risk categorization again was performed by a single observer, this time on three different days separated by 2-week “washout periods”.

The inter-run phase comprised the assay being performed once daily on three separate, non-consecutive days by a single operator, and risk categorization yet again being made by one observer. The inter-operator phase involved three different operators, each conducting one full assay run, and once more, a single observer performing risk classification. Lastly, the inter-observer phase entailed one full run being conducted by a single operator, and risk categorization this time being made by three different observers.

Four authors (SS, PS, PN, AS) served as operators or observers during the different assay validation phases. An additional independent expert (Anette Gomolka, a dermatologist and dermatohistopathologist) also served as an observer.

The 7-marker staining performed during the first run of the intra-run/intra-operator phase of the study was re-used (“recycled”) as the first run for the next four validation phases (Figure 3). This “recycling” approach was used to decrease the need for slides and hence, the demand for tissue, without affecting the study comparisons. The need was decreased by a total of 560 slides, i.e., one slide for each of the signature’s 7 markers per sample × 20 samples for each of the four subsequent validation phases.

Altogether, 7 different assay runs were performed, resulting in a total of 980 stainings for the signature’s biomarkers (7 stainings × 7 assay runs for each sample × 20 samples), not including the 140 additional stainings (7 for each sample × 20 samples) to obtain the reference risk classification.

### 2.2. Study Endpoints

The study’s primary endpoint was concordance, reflected by Fleiss’ kappa [19], between pairs of 7-marker signature assay risk category assignments (high-risk or low-risk) made within each assay validation phase. According to the CAP recommendations for analytical validation of immunohistochemical assays [17], a concordance of ≥0.9 demonstrates repeatability and reproducibility. The secondary endpoint was the percent agreement between the classifications made in each validation phase and the corresponding reference classificatons.

The study also had two exploratory endpoints: (1) comparison via box plot of the 7-marker signature assay overall risk scores and risk classifications among the five study phases and seven study runs versus the risk scores and risk classifications made in the reference staining, and (2) comparison of the overall risk scores in each study phase and run versus the reference overall risk score, via the creation of Bland–Altman plots.

### 2.3. Inclusion/Exclusion Criteria and Ethics

For a tissue block to be eligible for the analysis, that sample had to contain primary cutaneous melanoma, as verified by positive S-100 staining performed upon receipt of the specimen at the study laboratory; the original melanoma diagnosis had to have occurred between 2008 and 2021. An additional eligibility requirement was that a sample contain a sufficiently thick tumor to prepare the total of 50 evaluable slides, each 4 μm thick, that were required for the study staining. The 50 slides comprised one slide for the initial confirmatory S-100 test, and 49 slides, i.e., one slide to stain for each of the signature’s 7 biomarkers, for each of the study’s 7 runs (Figure 3).

Because of this tumor thickness requirement, samples had to consist of AJCC v8 T2 or T3 tumor tissue, as histologically confirmed by the respective study center. Specimens could not consist of the AJCC v8 T1 tumor, because, while thick enough for routine 7-marker testing, such tumor samples were not expected to be able to provide the much higher number of slides needed for this study. AJCC v8 T4 tumor was not analyzed, since this T-stage does not occur in AJCC v8 stage I-IIA disease, which is currently the main intended setting for use of the 7-marker signature assay.

Exclusion criteria were tumor tissue derived from melanoma re-excision, S-100 negativity in the initial study staining, or both.

A project description of the study was submitted for ethics committee review by each participating center, and work on the study was initiated after local ethics approval (see the Institutional Review Board Statement for additional details). Patients’ written consent for use of their excised and stored tissue for scientific research and scientific publications was obtained as required by local regulations and/or the local ethics committee.

### 2.4. Sample Quality Control and 7-Marker Testing

After FFPE tissue blocks were received at the central laboratory, each block underwent a microscopic and macroscopic quality control examination by an experienced laboratory scientist. Specimens were excluded from the study if, due to suboptimal preparation or storage, or for other reasons, the blocks showed physical defects that could lead to inevaluable 7-marker staining results. These defects included cracks in the tissue or paraffin, dryness, or flaking. Specimens also were excluded at this stage if they lacked sufficient tumor for all of the study’s testing. To assess whether a specimen had sufficient tumor, an S-100 test was performed on the first slide and an additional S-100 test was performed at a second timepoint during the subsequent testing. Both S-100 tests had to show a sufficient area of positive S100 staining for the study 7-marker testing to be completed. FFPE blocks with positive S-100 staining that passed the quality control examination underwent the reference 7-marker signature testing, and any such subsequent testing, according to standard operating procedures [12]. Briefly, for each melanoma specimen, staining for each of the 7 markers (representative images in Figure 4) was carried out using a proprietary panel of commercially available antibodies (Appendix A). The manufacturers provided information about the sensitivity and specificity of each antibody for its respective ligand. All antibodies were applied in ready-to-use formulations or standard dilutions. The same antibody solutions were used for all staining runs, and all stainings were performed within the pre-planned period, the 14 days after slide preparation. For all staining runs, reagents were prepared from single lots, and used within their expiry dates.

For each of the seven markers, a single score was determined based on a combination of the percentage of cells that stained for the marker, and the intensity of the immunoreactivity. This score was determined based on a 4-point stepwise scale ranging from 0, no staining, to 3, very strong staining. For each marker, this range of scoring was based on the percentage of staining and the intensity of immunoreactivity that are typically seen for that particular marker, i.e., scoring was individualized to each marker.

Each marker’s score was multiplied by the respective marker’s co-efficient, which had been derived in the 7-marker signature discovery study [12]. The products then were summed and normalized by the total number of markers, in this case 7, assessed. Thereby, an overall risk score was calculated ranging from −0.226 to +0.521. Overall scores were dichotomized into 7-marker signature assay high-risk or low-risk classifications, with scores >0.135 denoting the former. This threshold was determined in the initial 7-marker signature discovery study [12] and has been used in the validation cohort in that study, and in all subsequent published clinical validation studies [13,14,20].

### 2.5. Statistics

Statistical analysis was performed by Staburo GmbH, Munich, Germany, using R software version 4.0.3 (R Foundation for Statistical Computing, Vienna, Austria). No sample size was formally calculated. Instead, the sample size was based on CAP recommendations regarding analytical validation of nonpredictive, e.g., prognostic, immunohistochemical assays [17]. These recommendations suggest the use of 10 samples for each possible assay classification (in the case of the 7-marker signature, two classifications, high-risk or low-risk). Hence, the study was to include 10 high-risk samples and 10 low-risk samples, i.e., altogether 20 specimens from discrete patients. To increase the representativeness of the melanoma samples, the protocol specified that half of these specimens were to come from patients with AJCC v8 T2 disease, and half from patients with T3 melanoma.

Based on clinical validation data [12,13,14], the ratio of 7-marker signature high-risk to low-risk classifications in AJCC v8 T2/T3 melanomas was assumed to be 3:2, i.e., 60% of T2/T3 samples were assumed to be high-risk and 40%, low-risk. Based on these assumptions and on the estimated rate of samples staining negative for S-100 or failing the quality control examination, 28 tissue blocks were expected to be needed for screening at the central laboratory. This number of specimens would allow the inclusion of at least 10 evaluable samples from each risk category, half from each of the two T-stage subgroups in the study (Appendix A).

Pairwise agreement, i.e., concordance between pairs of risk classifications, was determined using Fleiss’ kappa [19] (primary endpoint) and percent agreement (secondary endpoint). Percent agreement was investigated based on positive percent agreement, negative percent agreement, and overall percent agreement, as recommended by CLSI I/LA28-A2 Guidelines [18]. Positive percent agreement was defined as the number of concordant low-risk classifications in a given study phase divided by the number of low-risk classifications according to the reference staining. Negative percent agreement represented the number of concordant high-risk classifications in a given study phase divided by the number of high-risk classifications according to the reference staining. Overall percent agreement was defined as the number of concordant classifications divided by the total number of comparisons in the given study phase. For all percent agreement values, 95% confidence intervals (CIs) were calculated according to the method of Clopper and Pearson [21].

## 3. Results

As planned, tissue blocks from 20 discrete patients were included. Of these, 12 blocks (T2 *n* = 6, T3 *n* = 6) were classified as high-risk in the 7-marker signature assay reference staining, and 8 blocks (T2 *n* = 4, T3 *n* = 4), as low-risk. This distribution of risk classifications resulted from 2/10 initially selected low-risk blocks turning out to lack sufficient S-100-positive tissue to provide slides that would be evaluable for all study testing; thus, this pair of blocks was replaced by high-risk specimens that had sufficient such tissue. The high-risk specimens had been successfully screened, but were not originally included in the study, because the planned number of eligible high-risk specimens already had been reached (Figure 5).

Seven-marker signature risk classification was highly consistent across all analytical validation phases, as demonstrated by Fleiss’ kappas (Table 2) and overall percent agreement (Table 3). As reflected by Fleiss’ kappas, pairwise concordance in classification exceeded CAP’s recommended 0.9 level in 4/5 analytical validation phases (Fleiss’ kappas = 0.929–1.000), and closely approached that target in one phase (Fleiss’ kappa = 0.864 for the inter-operator phase). As reflected by the total overall percent agreement for all runs in the given analytical validation phases (96.7–100%), pairwise concordance exceeded the 0.90 target for all five phases. The same held true regarding the total positive percent agreement and total negative percent agreement for all runs in the respective phases (95.8–100% and 94.4–100%, respectively).

Box plots were created of the continuous risk scores grouped by their classification as high-risk or low-risk in the respective validation phase (Figure 6). Altogether, 300 risk scores—60 per validation phase—were analyzed. The box plots illustrate the high concordance demonstrated in the Fleiss’ kappa analysis. The box plots also depict the five deviations from the reference risk classification that were seen in the study. These deviations were observed in the (A) intra-run/intra-operator, (C) inter-run, (D) inter-operator, and (E) inter-observer validation phases. In four cases, one each in assay runs A3, D2, D3, and E3, the risk category was wrongly classified as low-risk, and in one case, in assay run C2, the risk category was wrongly classified as high-risk.

To further compare agreement between each staining in each validation phase and the reference staining, Bland–Altman plots were prepared (Figure 7). These plots demonstrated strong agreement on risk scores. For all validation phases, the mean of the difference in scores was ≤0.05, out of a maximum possible difference of 0.745, indicating the robustness of the 7-marker signature assay at the level of risk scores.

## 4. Discussion

In this study, we analytically validated an already clinically-validated immunohistochemical prognostic test, the 7-marker signature assay [12,13,14]. This test was designed to identify “hidden high-risk patients” in AJCC v8 early stages (IA-IIA) of cutaneous melanoma. The stage I–IIA patients flagged by this assay as “high-risk” appear to have comparable long-term recurrence and mortality rates to those of unselected stage IIB patients and unselected stage IIC patients [4,13,15]. This observation, together with recent favorable results in adjuvant therapy clinical studies in stage IIB and IIC patients [9], suggests that the 7-marker signature assay could serve to pinpoint stage I–IIA patients who potentially might benefit from trials of adjuvant therapy and other intensified disease management strategies. By improving patient risk stratification, the assay could facilitate the selection of appropriate patients for such trials. In turn, by enriching the trials’ study samples, the assay could help increase the likelihood of clinical benefit of this investigation, speeding conduct and lowering costs. Indeed, if these studies prove successful, the 7-marker signature assay could ultimately make an important contribution to treatment and monitoring choices in everyday practice. Potentially, clinical use of the test could also include increasing confidence that given patients are truly low-risk, possibly allowing them to avoid unneeded intensified management.

Notably, to analytically validate the 7-marker signature assay, we used evaluation criteria based on recommendations by CAP [17] and CLSI [18], leading professional societies in monitoring and advancing immunohistochemical laboratory testing. We focused on the binary risk classification provided by the signature, which is the assay’s key output in aiding clinical decision making.

Our results demonstrated a high consistency of the 7-marker signature risk classification across all analytical validation phases. These findings confirmed this test’s precision across assay runs, operators, and observers.

As reflected by our primary endpoint, Fleiss’ kappa, pairwise concordance in our study exceeded the 0.9 target recommended by CAP for four out of five of our analytical validation categories and closely approached that target for the remaining category (Fleiss’ kappa 0.864). Of interest, it has been suggested that a Fleiss’ kappa ≥0.80 denotes “almost perfect agreement” [19,22].

As reflected by our secondary endpoint, percent agreement, whether overall, positive, or negative, the total pairwise concordance for all runs in the respective analytical validation phases exceeded the 0.9 threshold in all cases. These results, too, demonstrate that the 7-marker signature is robust and reliable over repeated use by a variety of operators and observers.

The signature’s strong performance in the inter-observer validation phase (Fleiss’ kappa 0.931, overall percent agreement for all runs, 98.3%, only 1 deviation from the reference among 60 risk score calculations shown in the corresponding box plot) was noteworthy. The assay proved similarly robust in this validation phase as in the other phases, notwithstanding that subjectivity of pathologists’ ratings of staining, and variation in grading by multiple observers are sometimes considered to be the main potential weakness of immunohistochemical testing [23,24,25,26].

The sole case where classification concordance did not exceed the 0.9 threshold was the Fleiss’ kappa for the inter-operator phase, but at 0.864, this variable approached that threshold. Additionally, and reassuringly, this same phase comfortably exceeded 0.9 concordance regarding overall percent agreement (96.7%), positive percent agreement (100%), as well as negative percent agreement (94.4%). The observation of a Fleiss’ kappa slightly below the 0.9 level likely can be explained by the influence of individual operator variation in a manual staining process.

Certain limitations of our work merit comment. First, the study did not involve a large number of specimens. Nonetheless, the number recommended by CAP for validation of a non-predictive, e.g., prognostic, immunohistochemical test with either of two potential classifications as its output, 20 from discrete patients [17], was achieved. Additionally, this was a multicenter study with three sites contributing samples, increasing generalizability.

Second, per protocol, T1 tumors were not included, since they would have been too thin to provide the unusually high number of slides needed for a study entailing seven runs each involving the staining of seven markers. However, the clinical validation literature for the 7-marker signature, which contains numerous patients with T1 disease, offers no indication of differing assay performance in T1 versus T2/T3 patients [12,14,20].

Lastly, although the protocol envisioned the inclusion of 10 specimens for each of the 7-marker signature assay’s two risk classifications, the study ultimately included 12 high-risk and 8 low-risk specimens according to the reference classification. As noted earlier, the reason for this changed study sample composition was the substitution of two “extra” high-risk specimens with sufficient tumor for preparation of all needed slides for two low-risk specimens that turned out to lack sufficient tumor. This substitution preserved the planned overall study sample size at the 20 tissue blocks recommended by CAP, and with that, the robustness of the projected statistical analyses. As well, the study sample maintained an appreciable proportion of low-risk samples, while at the same time reflecting the larger proportion of 7-marker signature high-risk versus low-risk classifications seen in patients with stage IB or, especially, IIA disease [13,14]. These settings may be regarded as “new frontiers” for potential changes in melanoma management following the introduction of adjuvant treatment for stage IIB and IIC disease [9]. Thus, stages IB and IIA melanoma may be the settings in which the 7-marker signature assay would be of greatest interest to enhance the conduct of clinical studies, and, ultimately, if warranted by the results of such trials, to aid treatment decisions in everyday practice [13].

It should be noted that since the present work was an analytical validation study, we did not assess whether the risk classifications made here were reflected in patient outcomes. Nonetheless, previously published clinical validation studies suggest good prognostic performance of the 7-marker signature assay [12,13,14].

## 5. Conclusions

In conclusion, the high concordance of risk classifications across assay runs, over time, and across operators and observers that was seen in this study demonstrates the repeatability, reproducibility, and hence the precision of the 7-marker signature assay. Additionally, the large number of stainings in this study (in total, seven biomarkers each stained in seven different runs) provides an impressive backdrop for these observations. These analytical validation data, together with the previously reported clinical validation in >750 patients, suggest that the 7-marker signature is “fit-for-purpose” and of clinical utility to stratify patients with AJCC v8 early-stage melanoma into high-risk and low-risk categories regarding recurrence and disease-specific mortality. Additional clinical validation, e.g., in prospective, randomized trials of adjuvant therapy, ultimately will increase confidence in the application of this assay in the everyday care of patients with melanoma.

## Figures and Tables

**Figure 1 diagnostics-13-03096-f001:**
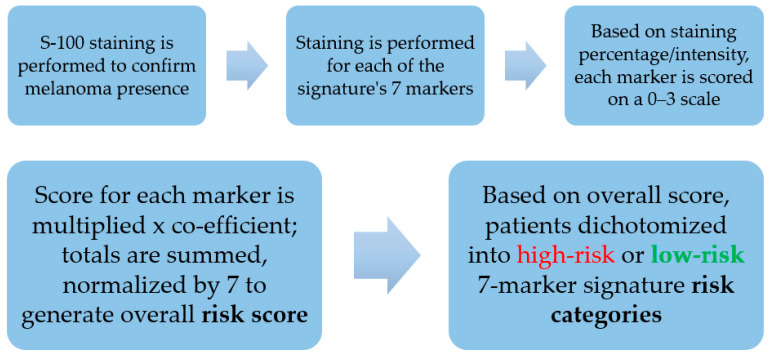
7-marker signature assay (immunoprint^®^) workflow. A proprietary panel of antibodies is used for staining that measures the expression of the 7-marker signature’s five “risk markers” (Bax, Bcl-X, COX-2, PTEN, presence of CD20-positive B-lymphocytes) and two “protective markers” (loss of MTAP and of ß-catenin). Co-efficients used to calculate the overall 7-marker signature score were determined in the assay’s development and initial clinical validation study, as was the threshold, >0.135 on a −0.226 to +0.521 scale, that delineates high-risk versus low-risk categories [12].

**Figure 2 diagnostics-13-03096-f002:**
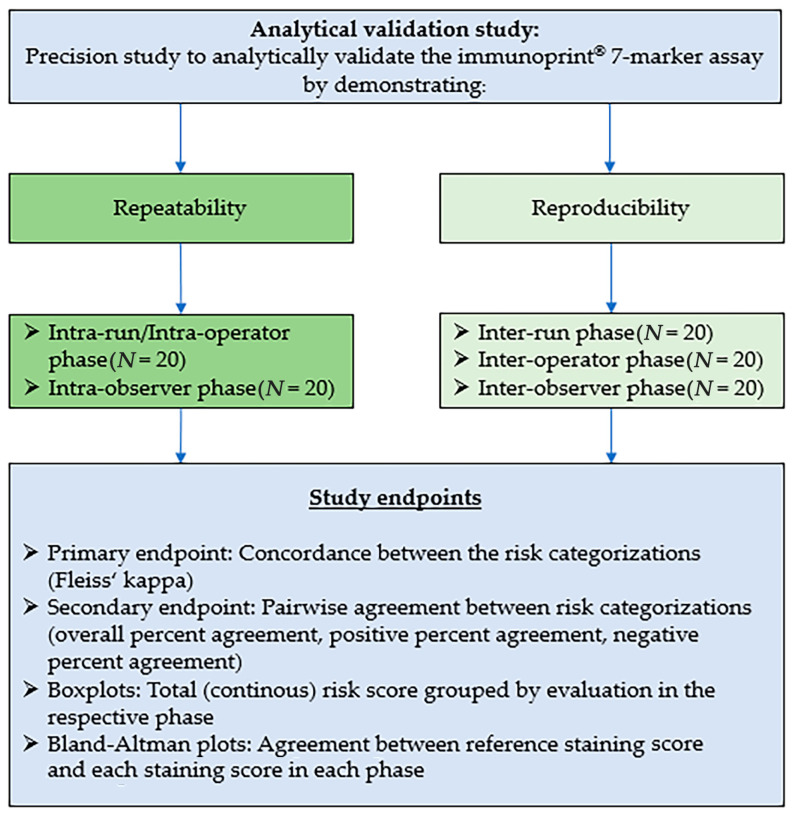
Summary of analytical validation study goals, endpoints, and design.

**Figure 3 diagnostics-13-03096-f003:**
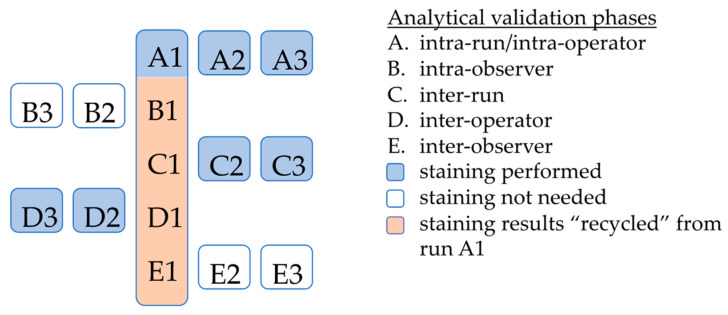
The study’s 7-marker signature assay (immunoprint^®^) staining and analytical validation scheme. The study scheme comprised 5 analytical validation phases entailing a total of 7 assay runs. The initial run of the first validation phase (run A1, in blue), i.e., the intra-run, intra-operator phase, was re-used (“recycled”) as the first run of each of the next four analytical validation phases, to decrease the need for melanoma tissue/slides. Each analytical validation phase included 3 risk categorizations for each of the 20 samples, or a total of 60 risk categorizations. These were compared with a separate “reference risk classification” made before the analytical validation phases started. The numbers after the letters in the figure refer to the assay run number in the given analytical validation phase, e.g., B3 denotes the third assay run in the intra-observer phase.

**Figure 4 diagnostics-13-03096-f004:**
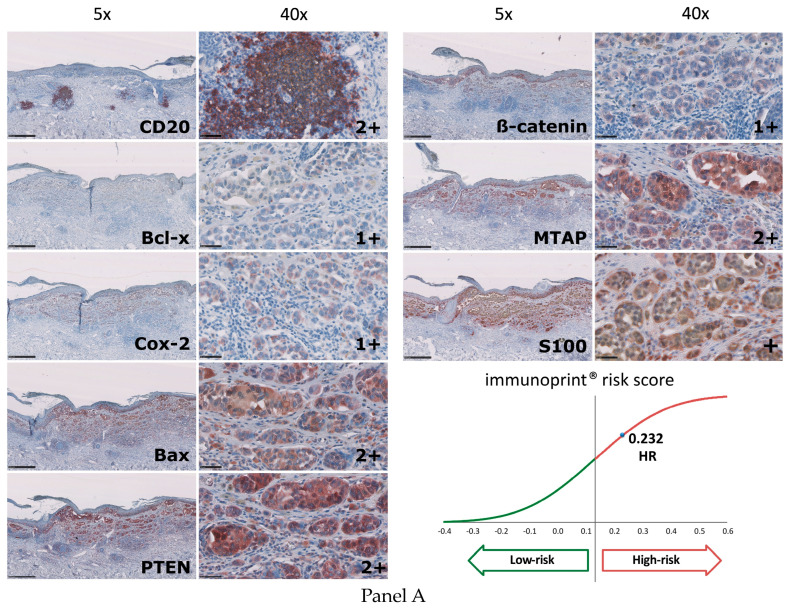
Representative images of study staining for the biomarkers in the 7-marker signature and for the S-100 marker of melanoma. The degree of magnification appears above each column of images. Marker names appear in the lower right-hand corner of the respective image in the left-most column. For each of the signature’s seven markers, a score is assigned on a 4-point stepwise scale (0–3) that increases along with the percentage and intensity of staining. Note that for each marker, this range of scoring is based on the percentage of staining and intensity of immunoreactivity that are typically seen for that particular marker, i.e., scoring is individualized to each marker. The score for the given marker appears in the lower right-hand corner of the respective image in the right-most column. The overall 7-marker risk score appears in the lower right-hand corner of each panel. Panel A: high-risk patient and Panel B: low-risk patient, both with AJCC v8 T3b primary tumor classification. Scale: 500 µm for the 5× magnification, 50 µm for the 40× magnification.

**Figure 5 diagnostics-13-03096-f005:**
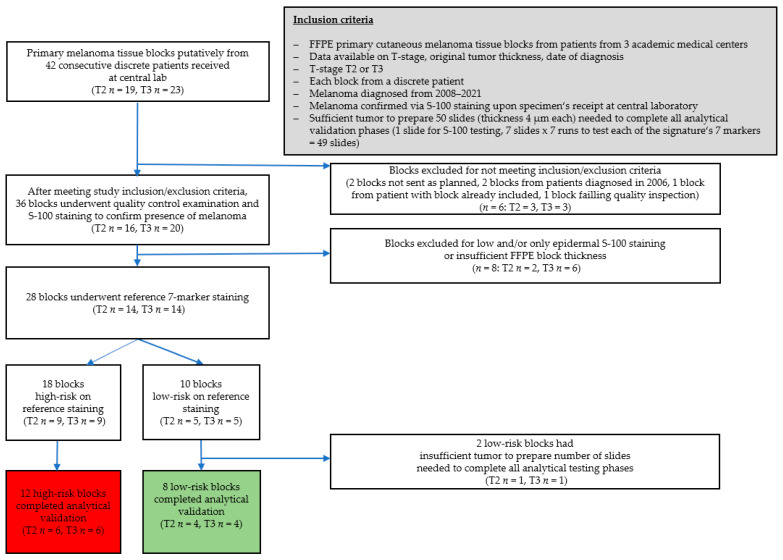
Specimen inclusion flow. The final study sample comprised 20 primary cutaneous melanoma tissue blocks, with melanoma confirmed by positive staining for S-100, from 1 discrete patient per specimen. Ten of the specimens were from patients with stage T2 primary tumor and ten were from patients with stage T3 primary tumor, per protocol. Of the 20 blocks, 8 blocks were 7-marker signature low-risk, and 12 blocks, 7-marker signature high-risk, in reference testing. The change from the planned 10 specimens per risk category was due to substitution of 2 high-risk blocks with sufficient tumor to prepare all 49 slides needed to complete all analytical validation phases for 2 low-risk blocks. The latter blocks had passed screening and undergone reference staining, but turned out to lack sufficient tumor for the 49 slides. After the substitution, there were still the same number of T2 vs. T3 samples within each risk category, as originally envisioned. FFPE, formalin-fixed, paraffin embedded.

**Figure 6 diagnostics-13-03096-f006:**
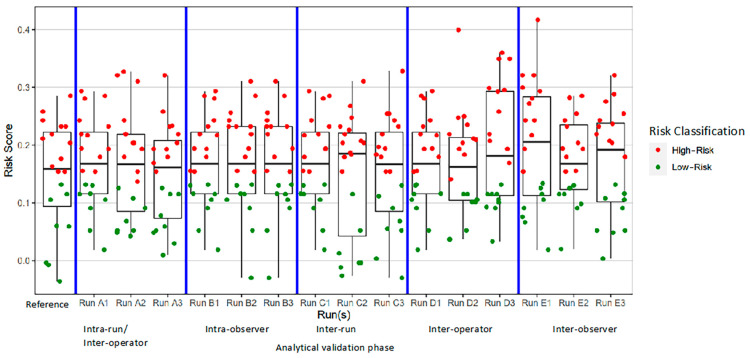
Box plot of 7-marker signature overall risk scores by analytical validation study phase and individual observation. Reference overall risk scoring and overall risk scoring in each of the five analytical validation study phases are separated by vertical blue lines. Horizontal gray lines denote the mean overall risk score for a run/set of observations, while gray rectangles encompass the standard deviation of that mean.

**Figure 7 diagnostics-13-03096-f007:**
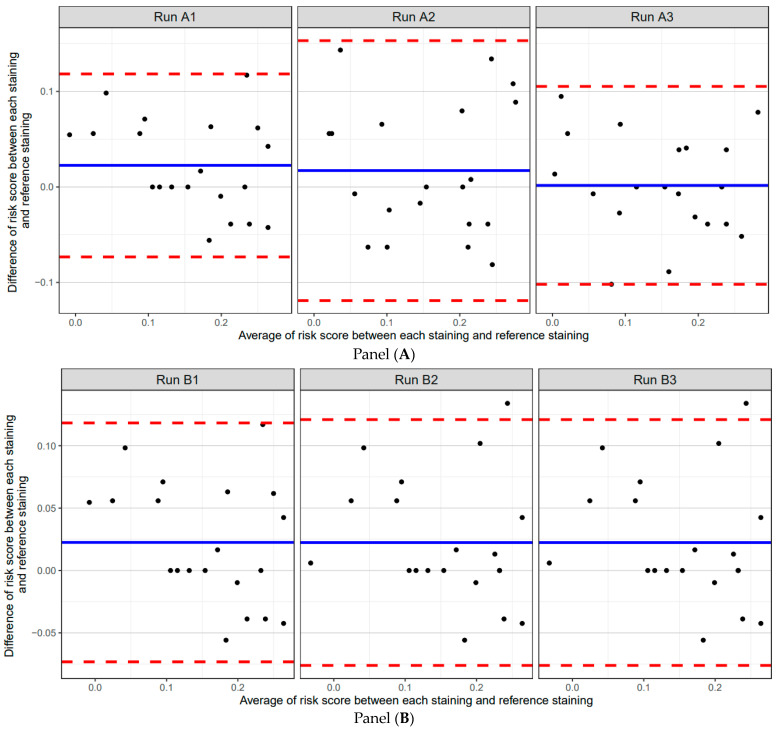
Bland–Altman plots of differences between analytical validation phase 7-marker overall risk scores and reference overall risk scores, by analytical validation phase. Black dots denote differences between the individual scores in a given analytical phase versus the reference scores for the same specimens. Solid blue lines denote the mean of the differences between the overall risk scores, while red dashed lines denote the standard deviation of that mean. Panel (**A**) intra-run/intra-operator phase, Panel (**B**) intra-observer phase, Panel (**C**) inter-run phase, Panel (**D**) inter-operator phase, and Panel (**E**) inter-observer phase.

**Table 1 diagnostics-13-03096-t001:** Design of validation study phases.

Validation Study Phase	Goal	No. of Full Assay Runs *	No. of Operators †	No. of Observers **	Derivation of Total Number of Observations per Phase ††	Comment
A. Intra-run/intra-operator	Establish repeatability	3	1	1	1 observation × 3 runs × 20 specimens = 60	
B. Intra-observer	Establish repeatability	1	1	1	3 observations × 1 run × 20 specimens = 60	Assessments performed separately, 2 weeks apart
C. Inter-run	Establish reproducibility	3	1	1	1 observation × 3 runs × 20 specimens = 60	One run per day performed on separate, non-consecutive days
D. Inter-operator	Establish reproducibility	3	3	1	1 observation × 3 runs × 20 specimens = 60	
E. Inter-observer	Establish reproducibility	1	1	3	3 observations × 1 run × 20 specimens = 60	

* To decrease the need for tumor tissue for slide preparation, the first run of the initial analytical validation phase was re-used (“recycled”) as the initial run of each of the next four phases. Hence a total of 7 runs, entailing a total of 140 sets of stainings (i.e., 7 runs each × 20 specimens) were performed in the study (Figure 2). Since each analytical validation study phase included 60 observations/categorizations, the study included a total of 300 risk categorizations (i.e., 5 phases × 60 observations/categorizations each). † “Operators” refers to laboratory scientists; ** “Observers” refers to dermatopathologists or other scientists trained in the 7-marker signature assay methodology; †† “Observations” refers to 7-marker signature risk categorizations.

**Table 2 diagnostics-13-03096-t002:** Concordance in 7-marker signature risk classification between pairs of classifications in the respective analytical validation phases, by phase.

Type of Potential Variation/Analytical Validation Phase	Fleiss’ kappa
(A) Intra-run/intra-operator *	0.931
(B) Intra-observer †	1.000
(C) Inter-run	0.929
(D) Inter-operator	0.864
(E) Inter-observer	0.931

* ”Operator” refers to a laboratory scientist; † “Observer” refers to a dermatopathologist or other scientist trained in the 7-marker signature assay methodology.

**Table 3 diagnostics-13-03096-t003:** Percent agreement with the reference 7-marker signature risk classifications by run and analytical validation phase.

Type of Potential Variation/ Analytical Validation Phase	Run	Percent Agreement, % (95% CI)		
		Positive	Negative	Overall
(A) Intra-run/intra-operator *	Run A1 †	100% (63.1–100%)	100% (73.5–100%)	100% (83.2–100%)
	Run A2	100% (63.1–100%)	100% (73.5–100%)	100% (83.2–100%)
	Run A3	100% (63.1–100%)	91.7% (61.5–99.8%)	95.0% (75.1–99.9%)
	All runs in phase A	100% (85.8–100%)	97.2% (85.5–99.9%)	98.3% (91.1–100%)
(B) Intra-observer **	Run B1 †	100% (63.1–100%)	100% (73.5–100%)	100% (83.2–100%)
	Run B2	100% (63.1–100%)	100% (73.5–100%)	100% (83.2–100%)
	Run B3	100% (63.1–100%)	100% (73.5–100%)	100% (83.2–100%)
	All runs in phase B	100% (85.8–100%)	100% (90.3–100%)	100% (94.0–100%)
(C) Inter-run	Run C1 †	100% (63.1–100%)	100% (73.5–100%)	100% (83.2–100%)
	Run C2	87.5% (47.3–99.7%)	100% (73.5–100%)	95.0% (75.1–99.9%)
	Run C3	100% (63.1–100%)	100% (73.5–100%)	100% (83.2–100%)
	All runs in phase C	95.8% (78.9–99.9%)	100% (90.3–100%)	98.3% (91.1–100%)
(D) Inter-operator	Run D1 †	100% (63.1–100%)	100% (73.5–100%)	100% (83.2–100%)
	Run D2	100% (63.1–100%)	91.7% (61.5–99.8%)	95.0% (75.1–99.9%)
	Run D3	100% (63.1–100%)	91.7% (61.5–99.8%)	95.0% (75.1–99.9%)
	All runs in phase D	100% (85.8–100%)	94.4% (81.3–99.3%)	96.7% (88.5–99.6%)
(E) Inter-observer	Run E1 †	100% (63.1–100%)	100% (73.5–100%)	100% (83.2–100%)
	Run E2	100% (63.1–100%)	100% (73.5–100%)	100% (83.2–100%)
	Run E3	100% (63.1–100%)	91.7% (61.5–99.8%)	95.0% (75.1–99.9%)
	All runs in phase E	100% (85.8–100%)	97.2% (85.5–99.9%)	98.3% (91.1–100%)

Each phase included specimens from 20 discrete patients and involved 60 observations (i.e., risk categorizations). In each phase, each 7-marker signature risk categorization for a run or by an observer was compared with the reference categorization for the specimen. The reference categorization was determined independently from categorizations in the analytical validation phases, after positive S-100 staining confirmed the presence of melanoma in a specimen, and after that specimen passed the protocol-specified quality control inspection. Overall percent agreement was calculated by dividing the number of concordant classifications by 20, the total number of specimens. Positive percent agreement was calculated by dividing the number of concordant low-risk classifications by the total number of low-risk classifications, then multiplying by 100. Negative percent agreement was calculated by dividing the number of concordant high-risk classifications by the total number of high-risk classifications, then multiplying by 100; CI, confidence interval; ***** “Operator” refers to a laboratory scientist; † To decrease the need for tumor tissue for slide preparation, run A1, i.e., the initial run of the intra-run/intra-operator analytical validation phase, was re-used (“recycled”) as the initial run of each of the next four analytical validation phases (Figure 3); ****** “Observer” refers to a dermatopathologist or other scientist trained in the 7-marker signature assay methodology.

## Data Availability

All relevant data are within the manuscript and its supporting tables and figures.

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
