# Peer review of "Analytical Validation of an Immunohistochemical 7-Biomarker Prognostic Assay (immunoprint®) for Early-Stage Cutaneous Melanoma in Archival Tissue of Patients with AJCC v8 T2–T3 Disease"

_diagnostics, 2023, doi:10.3390/diagnostics13193096_

Round 1

Reviewer 1 Report (Previous Reviewer 2)

The authors have addressed the previous comments. The updated figures are much more representative and have the correct labeling

Reviewer 2 Report (Previous Reviewer 3)

The manuscript has been improved.

This manuscript is a resubmission of an earlier submission. The following is a list of the peer review reports and author responses from that submission.

Round 1

Reviewer 1 Report

The article is well written, the data are original and interesting, the methods are correct and the conclusion are well supported by the results presented. 

The article is well written, the data are original and interesting, the methods are correct and the conclusion are well supported by the results presented. 

I suggest the paper should be published

Reviewer 2 Report

This is a well-written study titled "Analytical Validation of an Immunohistochemical Seven-Biomarker Prognostic Assay (immunoprint®) for Early-Stage Cutaneous Melanoma in Archival Tissue of Patients with AJCC v8 T2–T3 Disease." The study investigates the reproducibility of using a seven-biomarker assay on primary melanoma samples from 20 patients. The authors show high consistency across validation categories. The recommendations for the authors are:

1. I acknowledge that the authors were blinded to the Breslow thickness of tumors, however, in the quality control section it is mentioned that "Specimens also were excluded at this stage if they lacked sufficient material for all of the study's testing," can you please clarify what is the minimum requirement for testing? is there a minimal number of cells that need to be met to consider the test valid (e.g., PD-L1 IHC usually needs 100 cells to consider the sample eligible for testing. If you can add a short paragraph on the minimum requirements for testing that would be beneficial for the reader.

2. Figure 4 needs to be fixed. The histopathologic sections are inverted (skin surface to the bottom) and there is an obvious mislabeling for the magnification, if they are all at 10X magnification then they should all be the same size; however, some appear at 10X and another set appears at 20X most likely. please correct the figure and please attempt to obtain publication-quality images of the IHC. 

Reviewer 3 Report

The authors have shown in this paper that the 7-marker signature assay, a test to predict the risk of recurrence in patients with relatively early stage T2-3 (stage IB-IIB, AJCC 8th) thickness melanoma, has no variability in results between test assessors.

Although the number of samples used in this report is limited, the lack of variability in results indicates that this test is excellent for assessing low and high risk groups on the 7-marker signature assay.

However, because there is no quality control of whether the results of the classification of hign risk and low risk groups in this study accurately distinguish between high and low risk groups in melanoma recurrence, the results of this study do not provide an accurate picture of the risk of recurrence in patients with early-stage melanoma.

If the authors conducted the current study using a portion of the sample used in previous cohorts, such as study by Meyer et al. or by Garbe et al., this should be mentioned and shown to be highly consistent with previous studies.

If long-term recurrence was not assessed in the sample used in the current study, then the authors should explicitly mention the possibility that the results of the current study do not necessarily correlate with the results of the long-term recurrence rate.

Reviewer 4 Report

Dear editor and authors of the article:

The article entitled "Analytical Validation of an Immunohistochemical Seven-Biomarker Prognostic Assay (immunoprint®) for Early-Stage Cutaneous Melanoma in Archival Tissue of Patients with AJCC v8 T2–T3 Disease" is of interest as a preliminary study that demonstrates the usefulness of a battery of specific antibodies in the classification and detection of cutaneous melanoma, although it must be investigated in more depth to be able to be applied in the clinic; this must be specified at all times.

A series of suggestions for improving the manuscript are proposed below:

1. The abstract is incomplete. It is suggested to make a paragraph that includes information about the introduction, material and methods, results and discussion, since one must not forget to introduce the topic to be developed.

2. Results: the IHC images should be clearer, since the immunoreaction is not observed accurately, since most of them do not present enough contrast to be considered positive. In addition, a non-specific immunoreaction seems to be observed in some of them, so it is suggested to improve the quality of the images, and even take some examples at higher magnifications to see the cellular distribution of the antibodies. The legend of the images must include the explanation of all the data provided in the images, even if it is mentioned in the text (1+, 2+...).

The inclusion of the diagrams that can be seen throughout the manuscript that explain the procedure is greatly appreciated; They are useful for monitoring the process.

3. Table S1, of supplementary material, which includes the antibodies used in the study, should contain the following sections for your better knowledge: antigen/clone, origin (mouse, rabbit...), dilution and supllier/location.

Thank you so much.

Try to be more concise in the methodology and results